# 1000 Families Study, a UK multiwave cohort investigating the well-being of families of children with intellectual disabilities: cohort profile

Richard P Hastings [1,2] Vasiliki Totsika [1,2,3] Nikita K Hayden [1]
Caitlin A Murray [1] Mikeda Jess,[1] Emma Langley [1]
Jane Kerry Margetson [1,4]

¹Centre for Educational Development, Appraisal and Research (CEDAR), University of Warwick, Coventry, UK
²Centre for Developmental Psychiatry and Psychology, Department of Psychiatry, Monash University, Melbourne, Victoria, Australia
³Division of Psychiatry, University College London, London, UK
⁴Cerebra, Carmarthen, UK

**Correspondence to**
Caitlin A Murray;
C.Murray.7@warwick.ac.uk

## ABSTRACT

**Purpose** The 1000 Families Study is a large, UK-based, cohort of families of children with intellectual disability (ID). The main use of the cohort data will be to describe and explore correlates of the well-being of families of children with ID, including parents and siblings, using cross-sectional and (eventually) longitudinal analyses. The present cohort profile intends to describe the achieved cohort.

**Participants** Over 1000 families of UK children with ID aged between 4 and 15 years 11 months (total n=1184) have been recruited. The mean age of the cohort was 9.01 years old. The cohort includes more boys (61.8%) than girls (27.0%; missing 11.1%). Parents reported that 45.5% (n=539) of the children have autism. Most respondents were a female primary caregiver (84.9%), and 78.0% were the biological mother of the cohort child with ID. The largest ethnic group for primary caregivers was White British (78.5%), over half were married and living with their partner (53.3%) and 39.3% were educated to degree level.

**Findings to date** Data were collected on family, parental and child well-being, as well as demographic information. Wave 1 data collection took place between November 2015 and January 2017, primarily through online questionnaires. Telephone interviews were also completed by 644 primary caregivers.

**Future plans** Wave 2 data collection is ongoing and the research team will continue following up these families in subsequent waves, subject to funding availability. Results will be used to inform policy and practice on family and child well-being in families of children with ID. As this cohort profile aims to describe the cohort, future publications will explore relevant research questions and report key findings related to family well-being.

## INTRODUCTION

Intellectual disability (ID) is a condition described in International Classification of Diseases, 11th Revision (ICD-11) as a disorder of intellectual development.[1] Consistent with contemporary definitions of this condition, ID emerges during the 'developmental

---

### Strengths and limitations of this study

► A cohort of parental caregivers of children with intellectual disability (ID) to explore the well-being of children with ID, the well-being of parents and siblings, and the family system and relationships.
► The cohort includes over 1000 families and includes children across the spectrum of ID severity.
► Data will be collected longitudinally over a number of years, allowing investigation of family experiences over time.
► For the first two study waves, data are based on parental report only.
► The cohort is likely to mainly include those who have the time and motivation to participate.

---

period' (usually taken to mean before age 18 years), and is characterised by low cognitive ability (IQ<70) and low levels of adaptive behaviour (such as communication, social skills, independence skills—also assessed using standardised tools). Prevalence studies internationally suggest that approximately 3%–4% of children and adolescents have an ID.[2 3]

For several decades, researchers throughout the world have asked questions about whether the health (especially mental health) of family members of children with ID is different to that in families that are not raising a child with ID. Meta-analyses and results from higher quality research designs (such as population-based national samples) have suggested that mothers of children with ID are about 1.5 times more likely than other mothers to experience depression,[4] and similarly 1.4 times more likely than other mothers to have high levels of psychological symptoms indicative of mental health problems.[5] Data from the UK Millennium Cohort Study (a population representative birth cohort) also

show that fathers of young children with ID are twice as likely to score above the cut-off on a psychiatric disorder screen when compared with fathers of other young children[6] and older siblings of young children with ID are 1.5 times more likely to score in the 'abnormal' range on a screen for behavioural and emotional problems when compared with other siblings of young children.[7]

Sometimes explicitly, but often implicitly, researchers making these comparisons between families of children with ID and those without are drawing on a family systems perspective that has been applied to families of children with a variety of severe cognitive and developmental disabilities.[8–11] The core idea is that families are systems and so changes in one part of the family system, or in one family subsystem (eg, spousal, parent–child, sibling–sibling), will affect other family members and/or other family subsystems.

Families of children with ID face multiple challenges. For example, children are more likely to, by definition, have cognitive and adaptive skill deficits, and are four to five times more likely to have mental health problems compared with other children.[12] Families are also at increased risk for multiple social/economic disadvantage including poverty and negative life events.[12] The day-to-day burden of care for a child with ID is high even when compared with other carers (eg, dementia family carers),[13] but access to supports and services for children with ID and their families is limited or fraught with negative experiences often described as 'battling' against the system.[14 15]

Given these challenges facing families of children with ID, we might expect multiple negative effects on (for example) the psychological adjustment of parents and siblings. Research findings are generally consistent with this prediction. From a systems perspective, we would also expect family subsystems to be negatively affected. Again, research evidence supports this prediction. For example, using the population-based UK Millennium Cohort Study, we showed that parents of young children with ID reported more conflict and lower levels of closeness in the relationship with their child compared with other parents of young children.[16] In addition, meta-analytic reviews suggest that parents of children with disabilities, including ID, may be more likely to separate/divorce and have lower levels of satisfaction with their partner relationship compared with other parents.[17 18] We developed the 1000 Families Study to explore these issues further and to extend current knowledge across several domains. This article describes the cohort recruited for the 1000 Families Study. Future publications will concentrate on exploring relevant research questions and report key findings related to family well-being.

## Study aims

The main aims of the 1000 Families Study are as follows:
1. To describe the well-being of families of children with ID and to compare these data wherever possible with normative data and/or to national data sets (especially the UK Millennium Cohort Study). Family members include mothers (or primary parental caregivers), fathers (or secondary parental caregivers) and siblings.
2. To examine well-being differences between mothers and fathers in families of children with ID.
3. To explore correlates of maternal, paternal and sibling well-being and relationship quality, including deprivation and socioeconomic circumstances; other child, parent, sibling and family demographic factors; other family members' well-being and the child with ID's behaviour problems and adaptive skills.
4. To explore correlates of the behavioural and emotional well-being of the child with ID, including family deprivation; other child, parent, sibling and family demographic factors; other family members' well-being, the child with ID's adaptive skills, and parenting attitudes and behaviours.

## STUDY DESCRIPTION

To date, we have conducted an online cross-sectional survey and telephone interviews of over 1000 families of children with ID who were living in the UK at the time of recruitment. Recruitment involved a multipoint contact method using websites, social media, advertisements in disability organisations' newsletters, and by approaching special schools and parent support organisations. To examine longitudinal associations, we are currently inviting these same families to participate in further data collection (wave 2), which includes an online (or postal on request) survey but no telephone interview. We summarise the first and second waves of the 1000 Families Study.

### Wave 1 data collection

The primary (and secondary if available) parental caregiver were asked to complete the same online survey: 1184 primary caregivers and 93 secondary caregivers completed the wave 1 survey between November 2015 and January 2017. The primary caregiver was also invited to take part in a telephone interview following the questionnaire to provide information about the adaptive skills and behaviours of their child with ID. The telephone interview was arranged at a time convenient to the primary caregiver and most often took around an hour. The telephone interviews were completed by 644 primary caregivers. Differences between those who did and did not complete a telephone interview are discussed in the 'Findings to date' section.

### Wave 2 data collection

Participants for wave 2 are the caregivers who indicated during wave 1 that they were willing to be contacted again to be invited to participate in wave 2. The telephone interview was replaced with an additional measure on adaptive skills in the online survey. Caregivers are first contacted 2 years and 9 months after they completed wave 1. Wave 2 data collection began in August 2018, and is anticipated to end in January 2021.

## Study funding

The study is funded by Cerebra, a non-profit charity working with families with children with brain conditions. The project has also received funding from the Economic and Social Research Council Warwick Doctoral Training Centre, and the University of Warwick via PhD scholarship funding.

## Study population

### Inclusion criteria

1. For wave 1, families with one or more children aged between 4 and 15 years 11 months old who have an ID as reported by a parental caregiver. The child age range was initially (from November 2015) specified as 4 years to 11 years. In February 2017, an ethics amendment allowed us to expand recruitment to children aged up to 15 years and 11 months to meet families' interest in the study and to reach our sample target of over 1000 families.
2. Families with at least one parental caregiver responding to the survey.
3. Families who live within the UK (ie, England, Northern Ireland, Scotland or Wales).

### Exclusion criteria

1. For wave 1, families in which the child with ID lives outside of the family home on a full-time basis (eg, in a 52-week residential school placement).
2. Parental caregivers whose English literacy skills would not allow them to participate in an online survey and telephone interview.
3. Parental caregivers under 18 years old.

### Wave 1 cohort description

Tables 1 and 2 summarise the key sociodemographic variables of the cohort child and their family, respectively, at wave 1. All data were reported by the primary caregiver of the child with ID. These were usually the biological mother of the child with ID (78.0%) or another female relative (84.9% of the primary caregivers were female). Cohort children with ID were aged between 4 and 15 years and 11 months, with a mean age of 9.01 (SD=2.93). There were more boys than girls (61.8% were male; 27.0% were female; 11.1% missing). The vast majority of families lived in England (n=1031), with fewer living in Scotland (n=48), Wales (n=83) and Northern Ireland (n=9).

Table 1 provides some data indicative of ID severity in the cohort children. In terms of level of ID, 42.3% were reported by parents to have a severe-to-profound ID, and 42.2% a mild-to-moderate ID (15.5% missing). A mild-to-moderate ID was defined as 'children with a mild to moderate intellectual disability can typically communicate and look after themselves well, but may take a bit longer to learn new skills compared to other children of the same age'. A severe-to-profound ID was described as 'children with a severe to profound intellectual disability are likely to have complex and multiple difficulties which require extensive support to learn and carry out daily activities'.

**Table 1** Socioeconomic and demographic information for wave 1 Cerebra 1000 Families sample: child with ID

| Child | |
|---|---|
| Child age range (years) | 4.01–15.92 |
| Child age: mean (SD) (years) | 9.01 (2.93) |
| Missing age n | 175 (14.8%) |
| Child gender | |
| n of males | 732 (61.8%) |
| n of females | 320 (27.0%) |
| Missing gender n | 132 (11.1%) |
| Parental caregiver reported ID level | |
| Severe-to-profound ID | 501 (42.3%) |
| Mild-to-moderate ID | 500 (42.2%) |
| Missing information | 183 (15.5%) |
| Parental caregiver reported autism diagnosis of child | 539 (45.5%) |
| Down syndrome | 161 (13.6%) |
| Cerebral palsy | 87 (7.3%) |
| Child has a visual impairment | |
| Yes | 283 (23.9%) |
| No | 719 (60.7%) |
| Missing information | 182 (15.4%) |
| Child has a hearing impairment | |
| Yes | 164 (13.9%) |
| No | 847 (71.5%) |
| Missing information | 173 (14.6%) |
| Child has epilepsy | |
| Yes | 148 (12.5%) |
| No | 861 (72.7%) |
| Missing information | 175 (14.8%) |
| Child has mobility problems | |
| Yes | 584 (49.3%) |
| No | 422 (35.6%) |
| Missing information | 178 (15.0%) |
| Child has other physical health problems | |
| Yes | 485 (41%) |
| No | 523 (44.2%) |
| Missing information | 176 (14.9%) |
| Vineland adaptive behaviour composite | |
| Mean (SD) | 57.97 (11.14) |
| Range | 25–100 |
| Total responses | 640 (54.1%) |
| Missing information | 544 (45.9%) |
| Vineland communication standard score | |
| Mean (SD) | 62.02 (13.63) |
| Range | 25–117 |
| Total responses | 642 (54.2%) |
| Missing information | 542 (45.8%) |

Continued

**Table 1** Continued

| Vineland daily living skills standard score | |
|---|---|
| Mean (SD) | 57.53 (12.41) |
| Range | 25–103 |
| Total responses | 643 (54.3%) |
| Missing information | 541 (45.7%) |
| Vineland socialisation standard score | |
| Mean (SD) | 59.24 (12.40) |
| Range | 15–110 |
| Total responses | 643 (54.3%) |
| Missing information | 541 (45.7%) |
| Type of school | |
| n of children attending special school | 463 (39.1%) |
| n of children attending mainstream school | 333 (28.1%) |
| n of children attending mainstream school in either a special unit or special educational needs provision | 151 (12.8%) |
| Missing information | 178 (15.0%) |

ID, intellectual disability.

Inclusion policies in the UK schooling context means school type is not precisely indicative of children's needs, particularly in primary schooling, nonetheless, 39.1% of children were attending a special school, in addition to 12.8% attending a mainstream school in a special provision or unit, and 28.1% attending a mainstream school. The Vineland Adaptive Behaviour Composite scores ranged from 25 to 100, with the mean score being 57.97 (SD=11.13). The range of standard scores on the Vineland domains of communication (25–117; mean=62.02; SD=13.63), daily living skills (25–103; mean=57.53; SD=12.41) and socialisation (15–110; mean=59.24; SD=12.40) were similarly wide. Thus, the full range of ID severity is represented within this cohort.

Common co-occurring conditions in the cohort of children with ID included autism (45.5%) and Down syndrome (13.6%). Other diagnoses were less common, but included Fragile X, Angelman syndrome and numerous rare genetic conditions. A large number of children had additional physical health conditions or sensory impairments, including visual impairments (23.9%), hearing impairments (13.9%), epilepsy (12.5%), mobility problems (49.3%) and 'other physical health problems' (41%).

## Measures

The study captured information on parental positivity, parental mental health, care burden/stress, life satisfaction, parent relationship, family satisfaction, parenting, child behavioural and emotional well-being, child adaptive behaviour, sibling behavioural and emotional well-being and sibling relationship quality. Tables 3 and 4 summarise the study measures, with number of missing items for each variable during wave 1 indicated.

**Table 2** Socioeconomic and demographic information for wave 1 Cerebra 1000 Families sample: parent and family

| Primary caregiver and family | |
|---|---|
| Female primary caregiver | 1005 (84.9%) |
| Primary caregiver relationship to child | |
| Biological mother | 923 (78%) |
| Adoptive mother | 53 (4.5%) |
| Biological father | 46 (3.9%) |
| Grandmother | 13 (1.1%) |
| Missing information | 129 (10.9%) |
| Primary caregiver employment status | |
| In a job working for an employer | 391 (33.0%) |
| Looking after home and family | 388 (32.8%) |
| Self-employed | 103 (8.7%) |
| Doing something else | 100 (8.4%) |
| Missing information | 129 (10.9%) |
| Primary caregiver education level | |
| Degree level | 465 (39.3%) |
| Higher education but below degree level | 253 (21.4%) |
| A/AS levels or equivalent | 104 (8.8%) |
| 5 or more GCSEs at A*-C or equivalent | 74 (6.3%) |
| Some GCSEs passes or equivalent | 93 (7.9%) |
| No qualifications | 13 (1.1%) |
| Missing information | 179 (15.1%) |
| Primary caregiver relationship status | |
| Married and living with spouse/civil partner | 631 (53.3%) |
| Divorced/separated/single/widowed/not currently living with partner | 226 (19.1%) |
| Living with partner | 141 (11.9%) |
| Missing information | 186 (15.7%) |
| Primary caregiver ethnicity | |
| White British | 930 (78.5%) |
| White other (Irish, travelling community, other) | 55 (5.2%) |
| Asian/Asian British | 26 (2.3%) |
| Black (African/Caribbean/Black British) | 15 (1.3%) |
| Remaining ethnic groups (mixed/multiple ethnicity, Arabic, any other ethnic background, etc) | 23 (2.0%) |
| Missing information | 135 (11.4%) |
| Country | |
| England | 1031 (87.1%) |
| Scotland | 48 (4.1%) |
| Wales | 83 (7.0%) |
| Northern Ireland | 9 (0.76%) |
| Missing | 13 (1.10%) |
| Family living in most deprived 10% of neighbourhoods based on indices of multiple deprivation (IMD) | 90 (7.6%) |
| Missing IMD information | 34 (2.9%) |

Continued

| Table 2 Continued | |
|---|---|
| **Primary caregiver and family** | |
| Child with ID has a sibling aged between 4 and 15 years 11 months | |
| Yes | 612 (51.7%) |
| No | 360 (30.4%) |
| Missing information | 212 (17.9%) |
| Nearest-in-age sibling of child with ID has a longstanding illness, disability or infirmity | |
| Yes | 168 (27.5%) |
| No | 439 (71.4%) |
| Missing information (from those who indicated child has sibling in age range) | 5 (0.8%) |

A/AS levels, GCE Advanced level; GCSE, General Certificate of Secondary Education; ID, intellectual disability.

## Patient and public involvement

The survey content and method was designed collaboratively with Cerebra, an organisation that supports and advocates for families of children with developmental disabilities, including ID. The final author (JKM) participated in the design of the survey in her capacity as Cerebra staff member and PhD student at the University of Warwick. To date, the research team and Cerebra have collaborated to disseminate findings to families of children with ID through parent and practitioner research seminars that Cerebra regularly hosts.

## Ethics

The study is being conducted in accordance with the ethical principles of the British Psychological Society. There are no significant risks for the families from taking part in this study, but as some of the questions included in the survey ask about potentially challenging issues, such as parental and child well-being and difficulties, signposting to various resources and helplines has been provided in the information sheet and at the end of the survey for caregivers.

All participant information is stored in line with the requirements of the Data Protection Act 1998. While this study was approved and began before changes to the General Data Protection Regulation (GDPR) in 2018, data collection and procedures in wave 2 are in line with GDPR requirements. Caregivers are informed through information sheets and consent forms on what data are collected, why this is necessary, and how these data are stored. All aspects of the study are conducted on the basis of explicit written consent to take part in each stage of the study. All research data are stored electronically and password protected on firewalled University computers with access limited to the research team. The password and access to the research data, including participant information, is restricted to the Chief Investigator and the research team.

For wave 2 of the data collection, consent is collected as it was in wave 1 and participants are asked again if they would be willing to be contacted for any future research. At wave 2, a £10 gift voucher was introduced as a Thank You to caregivers who complete the survey. To mitigate against potential biases arising from the monetary incentive, it is made clear in the information sheet that caregivers do not have to answer any questions that make them feel uncomfortable and they will still receive the voucher.

## Dissemination

Throughout the study, results will be published in peer-reviewed academic journals. Presentations of study findings will be made at research conferences and seminars. The charity Cerebra and other stakeholders will be involved in methods of dissemination for reaching the wider community, particularly families of children with ID. Caregivers who took part in the study, and have consented, receive updates about the study through a newsletter and social media, which the public can also access. This allows findings to be summarised and presented in an accessible format in addition to any academic papers or presentations.

## COHORT DESCRIPTION

The remainder of this manuscript will provide a description of the cohort participating in the 1000 Families Study. No findings have yet been reported from the data on this cohort. Overall, the parent-reported severity of ID, educational setting and Vineland adaptive behaviour scale (VABS) scores indicate that this cohort includes children across a wide range of ID severity, although the VABS scores indicate more children in the mild-moderate range, but this would be expected given a lower prevalence for severe to profound ID.

Telephone interviews were completed by 644 of the primary caregivers indicating that families who completed both the survey and the telephone interview may have had access to more resources that enable them to contribute more time to research participation. For instance, deprivation levels may be associated with telephone interview participation. To identify family deprivation, we constructed a composite variable with values ranging from zero to four, incorporating measures of subjective poverty, hardship, income poverty and neighbourhood deprivation. Primary caregivers who did not complete telephone interviews had a mean deprivation composite score of 1.71 (SD=1.10) compared with a mean deprivation composite score of 1.37 (SD=1.09) for primary caregivers who completed a telephone interview ($t$(948) = 4.72; p<0.001; $d$=0.32; MD=0.35, 95% CI 0.20 to 0.49). Primary caregivers who completed the telephone interviews were more likely to have a higher education qualification (75.1%) compared with primary caregivers who did not complete a telephone interview (66.6%; $\chi^2$ (1, n=1002)=8.58; p=0.003; $\phi$ = 0.093). Primary caregivers who were single parents provided a telephone interview in 53.5% of cases, compared with primary

**Table 3** Parent and family measures used within the Cerebra 1000 Families Study

| Domain | Measure | Scoring | Measured at | Wave 1 measure completeness |
|---|---|---|---|---|
| Parental positivity | Positive gains scale (Jess, Bailey, Pit-ten Cate *et al*, Measurement invariance of the positive gains scale in families of children with and without disabilities) 7 items Perceived benefits for parent (5 items) What family has gained (2 items) | 5-point scale from 0 (*strongly agree*) to 4 (*strongly disagree*). Original scale had a lower score indicating higher positive gains. The scoring was reversed so that a higher score indicates higher positive gains. | W1, W2 | Complete n=1004 Missing n=180 |
| Parent mental health | Kessler 6 (K6)[36] 6 items | 5-point Likert scale asks caregivers how often they have experienced six symptoms over the last 30 days from 0 (*none of the time*) to 4 (*all of the time*). Total score is calculated by summing the responses. A score of 13 and above indicates serious mental illness.[37] | W1, W2 | Complete n=1000 Missing n=184 |
| Carer burden/stress | 'Impact of care-giving on carer' Survey of Informal Carers in Households 2009/2010[38] 7 items | Items ask individuals whether certain aspects of their lives have been affected by caring for another (*Yes* or *No*). Totals are generated for the responses, and higher scores indicate higher carer burden/stress. | W1, W2 | Complete n=963 Missing n=221 |
| Life satisfaction | Life satisfaction scale[39] 1 item | Single item measure that asks caregivers to rate their general life satisfaction from 1 (*completely dissatisfied*) to 10 (*completely satisfied*). Scores 6–10 indicate that respondents are 'satisfied overall'. Scores 4 and below indicate that participants are 'dissatisfied overall'. | W1, W2 | Complete n=998 Missing n=186 |
| Parent relationship— parent's perception of relationship with their partner (if applicable) | Disagreement over issues related to child[40] 1 item | This item asks caregivers to report on how often they disagree over issues relating to their child on a 6-point Likert scale (*Never* to *More than once a day*). There is also a response for 'Can't say'. | W1, W2 | Complete n=771 No live-in partner n=226 Missing n=187 |
| | Happiness of relationship scale[40] 1 item | Caregivers are asked to rate how happy they are in their relationship from 1 (*very unhappy*) to 7 (*very happy*). There is also a response for 'Can't say'. | W1, W2 | Complete n=771 No live-in partner n=226 Missing n=187 |
| Family functioning | Family APGAR scale[41] 5 items Adaptability, Partnership, Growth, Affection and Resolve (APGAR) | Respondents are required to rate the frequency of feeling satisfied with each parameter on a 3-point Likert scale, from 0 (*hardly ever*) to 2 (*almost always*). The scale is scored by summing the 5 items. A higher score indicates a greater degree of satisfaction with family functioning (8–10=*highly functional*, 4–7=*moderately dysfunctional*, 0–3=*dysfunctional*). | W1, W2 | Complete n=993 Missing n=191 |
| Sibling behavioural and emotional well-being | Strengths and Difficulties Questionnaire (SDQ) (Parent questionnaire)[42] 25 items Subscales: emotional problems, conduct problems, hyperactivity/inattention, peer relationship problems, prosocial behaviour | Caregivers are asked to assess the extent to which each statement applies to the child based on the last 6 months using a 3-point rating scale (0=*not* true to 2=*certainly* true). The items are divided into 5 subscales. A total difficulties score can be obtained by summing the first four subscale scores: *close to average* (0–13), *slightly raised* (14–16), *high* (17–19) and *very high* (20–40). | W1, W2 | Complete n=603 No sibling n=360 Missing n=221 |
| Sibling relationship | Sibling Relationship Questionnaire - Short Form (SRQ-SF) (Parent questionnaire) (revised)[43] 10 items Adapted version using key items from two subscales of SRQ–SF: warmth and closeness (6 items) conflict (4 items) | Caregivers are asked about the sibling relationship on a 5-point Likert scale from 1 (*very much*) to 5 (*extremely much*). These items can then be used to derive measures that encapsulate positive or negative aspects of the sibling relationship. | W1, W2 | SRQ warmth: Complete n=596 No sibling n=360 Missing n=228 SRQ conflict: Complete n=599 No sibling n=360 Missing n=225 |

Continued

**Table 3** Continued

| Domain | Measure | Scoring | Measured at | Wave 1 measure completeness |
|---|---|---|---|---|
| Parenting attitudes and behaviours | Child–parent relationship scale (CPRS)[44] 15 items Conflict (8 items) Closeness (7 items) | The caregiver states their feelings and beliefs about their relationship with their child and about the child's behaviour towards them, by responding to statements on a 5-point Likert scale (1=*Definitely does not apply* to 5=*Definitely applies*). CPRS generates a separate score for the individual conflicts and closeness constructs by summing the scores in each subscale | W1, W2 | Closeness: Complete n=978 Missing n=206 Conflict: Complete n=983 Missing n=201 |
| | Alabama Parenting questionnaire (Parent form)[45] 12 items Positive parenting (6 items) Inconsistent discipline (6 items) | Parents/caregivers are asked about their parenting behaviours on a five-point Likert scale (1=*Never* to 5=*Always*). The items in each scale are summed to obtain two scale scores. | W1, W2 | Positive parenting: Complete n=972 Missing n=212 Inconsistent discipline: Complete n=961 Missing n=223 |
| | Parent–child activities index 5 items | This questionnaire was constructed for the purposes of this study. Some of the questions have been used in national UK surveys such as the Millennium Cohort Study, and where relevant, items were adapted to be more appropriate for families of children with ID. The questions ask about parent–child shared activities. Answers are provided on a five-point Likert scale (from 1=*Not at all* to 5=*Every* day). | W1 *Removed from W2 due to parental feedback.* | Complete n=974 Missing n=210 |

W1 and W2 indicate wave 1 and wave 2.

caregivers from two-parent households, who provided a telephone interview in 65.3% of cases ($\chi^2$ (1, n=998)=10.30; p=0.001; $\phi$ = −0.102).

In addition to resources, differences between those who participated in telephone interviews and those who did not included differences in child profiles. Children of caregivers who did not take part in the telephone interview had higher total SDQ total behaviour scores with a mean score of 22.00 (SD=6.52) compared with a mean score of 20.63 (SD=6.48) for those who did participate ($t$(983) = 3.201; p=0.001; $d$=0.21; MD=1.38, 95% CI 0.53 to 2.22). Primary caregivers who did not complete the telephone interviews had children with ID who had fewer coexisting physical health problems (such as epilepsy, mobility problems, hearing or visual problems; mean=1.49; SD=1.22) compared with the children with ID of primary caregivers who did complete the telephone interviews (mean=1.76; SD=1.33; $t$(985) = −3.157; p=0.002; $d$=0.21; MD=−0.27, 95% CI −0.43 to −0.10).

Comparison with indicators from the 2011 census of England and Wales[19] indicate lower participation of families identifying as Black, Asian or Minority Ethnic (5.6% of the 1000 Families sample compared with 13% of the population of England and Wales), but expected numbers of families identifying as White British (78.5% of the 1000 Families sample, compared with 80.5% in the 2011 census) (see table 2 for full ethnicity information on the 1000 Families). In terms of educational qualifications, the survey sample were more highly educated (39.1% of the primary caregivers in the 1000 Families were educated to degree level, in comparison to 27% in the census).[20] While 23% of the population in England

and Wales had no educational qualifications, only 1.1% of the 1000 Families sample reported having no educational qualifications. These comparisons to the census data should be viewed with caution as the census data about qualifications include the full age-range of adults living in England and Wales in 2011, whereas our sample of primary caregivers disproportionately includes women of childrearing age across all four countries of the UK. Policy changes leading to the expansion of UK higher education in recent decades means we would expect that the primary caregivers in our sample were more likely to have completed degrees than were previous generations. Furthermore, the 2011 census data included people aged 16–18 and this group were less likely to have any qualifications yet, although they may have been in education. In terms of socioeconomic indicators, 7.6% of the families in the 1000 Families sample were living in the most deprived areas in their respective country based on the indices of multiple deprivation for England, Scotland, Wales and Northern Ireland.[21–24] These indices offer a measure of relative deprivation at a local scale by combining information on income poverty, education, unemployment, health, crime, barriers and access to services, housing and physical environment for each small area and then rank-ordering areas within each country.

### Future directions
The data will be analysed in cross-sectional and longitudinal studies. Where possible, data will be compared with data from normative or population-representative samples. Correlation and regression approaches will be

**Table 4** Child measures used within the Cerebra 1000 Families Study

| Domain | Measure | Scoring | Measured at | Wave 1 measure completeness |
|---|---|---|---|---|
| Child with ID behavioural and emotional well-being | Strengths and Difficulties Questionnaire (SDQ) (Parent questionnaire)[42] 25 items Subscales: emotional problems, conduct problems, hyperactivity/inattention, peer relationship problems, prosocial behaviour | Caregivers are asked to assess the extent to which each statement applies to the child based on the last 6 months using a 3-point rating scale (0=*not* true to 2=*certainly* true). The items are divided into 5 subscales. A total difficulties score can be obtained by adding the first four subscale scores and categorising them into a four band system: *close to average* (0–13), *slightly raised* (14–16), *high* (17–19) and *very high* (20–40). | W1, W2 | Complete n=985 Missing n=199 |
| Behavioural and emotional problems of the child with ID | Development behaviour checklist (parent version) [46] 96 items Telephone interview Subscales: disruptive/antisocial behaviour, self-absorbed, communication disturbance, anxiety, social relating | Items are scored on a 0, 1 or 2 rating scale (0=*Not true as far as you know*, 1=*Somewhat or sometimes true*, 2=*Very true or often true*). The Total Behaviour Problem Score is an overall measure of emotional and behavioural problems, and can detect clinically significant levels of overall emotional and behavioural disturbance (indicated by a score of 46 or greater). | W1 | Complete n=644 Did not consent to telephone interview n=198 Missing n=342 |
| Adaptive behaviour skills of the child with ID | Vineland adaptive behaviour scales—second edition (parent interview form) (VABS- II)[47] 30–40 min telephone interview Communication skills Daily living skills Socialisation skills | An overall composite score, and domain scores, can be derived with reference to age during typical development during which children can perform the task items. Four standardised scores in total were derived for the present research: adaptive behaviour composite, communication skills, daily living skills and socialisation skills. Age equivalency scores were also calculated. | W1 | Complete n=645 Did not consent to telephone interview n=198 Missing n=341 |
| Adaptive behaviour skills of the child with ID | The GO4KIDDS Brief Adaptive Scale[48] 9 items | Items cover communication, self-help skills, social interaction and support needs. Each item is rated on a 5-point scale where higher scores indicate greater skill level and greater independence. These item scores are summed to get an overall Adaptive Behaviour score. We have amended the original GO4KIDDS to include an additional item asking about alternative methods of communication following feedback from caregivers. | W2 Replaced the VABS-II interview | Not yet known |

W1 and W2 indicate wave 1 and wave 2.
ID, intellectual disabilities.

used to explore factors associated with the well-being of mothers, fathers, siblings and the child with ID. Where appropriate, models will account for the nested nature of data within families. The cohort provides scope to explore a range of research questions, for example, studies exploring associations between behavioural outcomes for children with ID and their siblings in relation to their sibling relationship quality, anxiety in children with ID with and without autism, and well-being in mothers of children with Down syndrome. The cross-sectional studies using data from wave 1 will also help to develop models and questions that can be explored through longitudinal research following future waves.

## STRENGTHS AND LIMITATIONS

This study is, as far as we are aware, the largest cohort of families with a child with ID in the UK and will make an important contribution to the research on the well-being of children with ID and their families. Particularly important is the longitudinal design which currently includes a second wave of data collection that is underway, with a third wave under development. This adds to the growing global research following families of children

with ID and/or autism over time. For example, the Adolescents and Adults with Autism (AAA) study, is an ongoing, longitudinal study with 406 participants, based in the USA, with its sample recruited from Massachusetts and Wisconsin.[25 26] A strength of the AAA study is that it has followed up families for over 20 years, beginning in 1998. Our study will differ in that we have recruited participants from across the UK, rather than specific geographic locations within a country. We have also recruited families of children with ID rather than autism specifically, although we do have a large group of children with autism within our sample (n=539). Selection of measures allows for some comparison with population-representative samples in the UK (eg, the Millennium Cohort Study[27]). Results will inform parents, clinicians and educational professionals on factors related to the well-being of children with ID, their parents and their siblings.

Designing this research study via a questionnaire and telephone interview allowed us to recruit a large sample from across the UK. However, this precluded an independent ascertainment of child ID, and/or associated developmental conditions (eg, autism). Instead, information on the child is based solely on parental report, although

this includes their educational setting, parent-reported severity of ID, any diagnoses and for those who took part in the telephone interview, a measure of child adaptive behaviour. The implication of this limitation is that the survey cannot verify the extent of clinical diagnoses available for each child with ID. Where research evidence has examined the validity of parental reports, it has suggested that parents' reports correspond well to clinical diagnoses available[28 29] and that parental reports on children's developmental outcomes are accurate.[30]

Although the sample size recruited at wave 1 is large (>1000 families), the sampling design (convenience sample recruited through social media and ID-related organisations) may have resulted in sampling bias, where families of children with ID in contact with charitable organisations and/or access to social media may be more likely to be included. However, although not fully representative of the UK, the findings discussed previously indicate that the 1000 Families sample includes elements of diversity that have the potential to provide insights into families with a child with ID from various backgrounds.

Another limitation of this study is that not all primary caregivers completed a telephone interview as well as the survey — 644 out of 1184 primary caregivers completed a telephone interview. As previously summarised in the 'Findings to date' section, there were demographic differences between these groups. For example, those primary caregivers who did not take part in the telephone interview had higher levels of poverty, lower levels of qualifications, were more likely to be a single parent and reported higher average SDQ total behaviour scores for their children with ID. The children of the primary caregivers who did not complete the telephone interviews had fewer coexisting physical health problem labels compared with the children of primary caregivers who did complete the telephone interviews.

Finally, although 772 of the primary caregivers in the sample indicated they were married and/or living with a partner, only 93 secondary caregivers participated in the survey. Overall low rates of participation of fathers or secondary caregivers in family research is a well-described phenomenon that may relate to role allocation between parents regarding child care, working commitments, and primary responsibility for contact with disability and educational services.[31–35] The 1000 Families Study directly recruited and encouraged fathers to take part in the study, but it is likely that a recruitment strategy primarily focused on the family was not specific enough to encourage greater participation from fathers and secondary caregivers. There is a need for further investigation on why other family members, particularly fathers, were not as likely to take part, and whether there are recruitment strategies which are more successful to reach fathers and secondary caregivers, as some research has highlighted the importance of inviting fathers directly and emphasising the importance of their participation.[31 35]

## COLLABORATION

Initial data analyses and publications will be generated by researchers at CEDAR, University of Warwick as part of the 1000 Families Study funded by Cerebra. However, the research team welcome researchers interested in future collaboration to contact the corresponding author with their expression of interest. Access and analysis of the data is currently only possible via the University of Warwick, due to the ethical approvals in place.

**Acknowledgements** We would like to thank all the families that participated in this study. We acknowledge the support from Cerebra in funding, developing and recruiting to this study; and the ESRC Doctoral Training Centre at the University of Warwick for funding support in the form of PhD studentships. We acknowledge the contributions of our study administrators, Alison Baker and Diana Smith. Various researchers assisted in conducting telephone interviews, alongside the authors, to collect data for wave 1 of this study and we acknowledge the support of Latoya Clarke, Farah Elahi, Gemma Gray, Liz Halstead, Sasha Mandair, Louise Rixon, Suzi Scott and Ruth Thomas.

**Contributors** RPH and VT conceptualised and codesigned the study, wrote the study protocol, contributed to the drafting of the manuscript for publication, reviewed and revised the manuscript and read and approved the final manuscript. NKH and CAM contributed to the design of the study, contributed to the drafting of the manuscript for publication, reviewed and revised the manuscript, and read and approved the final manuscript. MJ, EL and JKM contributed to the design of the study, contributed to the original protocol and read and approved the final manuscript.

**Funding** The authors acknowledge the funding support of Cerebra, UK, the Economic and Social Research Council Warwick Doctoral Training Centre, and the University of Warwick.

**Competing interests** None declared.

**Patient consent for publication** Not required.

**Ethics approval** Full ethical approval was granted for this study from the UK National Health Service (NHS) West Midlands—South Birmingham Research Ethics Committee: REC reference number: 15/WM/0267 (11 September 2015). An amendment was granted to extend the age range (7 March 2017). An amendment was granted for changes made to wave 2 (8 February 2018) and the ethics committee was informed of further minor changes made to the documents for wave 2 (27 February 2018). Sponsor approval was also obtained from the University of Warwick.

**Provenance and peer review** Not commissioned; externally peer reviewed.

**Data availability statement** No data are available. Data from this study are not available for sharing due to ethical approval requirements. Researchers interested in collaboration should contact the corresponding author with their expression of interest.

**ORCID iDs**
Richard P Hastings http://orcid.org/0000-0002-0495-8270
Vasiliki Totsika http://orcid.org/0000-0003-1702-2727
Nikita K Hayden http://orcid.org/0000-0003-1104-3885
Caitlin A Murray http://orcid.org/0000-0002-7547-736X
Emma Langley http://orcid.org/0000-0001-6311-1734
Jane Kerry Margetson http://orcid.org/0000-0003-3068-2076

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
