## [Reviewer comments · BMJ Open]

ARTICLE DETAILS

TITLE (PROVISIONAL)	Cohort Profile: The 1,000 Families Study, a UK multi-wave cohort investigating the wellbeing of families of children with intellectual disabilities.
AUTHORS	Hastings, Richard P; Totsika, Vasiliki; Hayden, Nikita K; Murray, Caitlin A; Jess, Mikeda; Langley, Emma; Margetson, Jane Kerry

VERSION 1 – REVIEW

REVIEWER	Lauren Taylor King's College London, Institute of Psychiatry, Psychology and Neuroscience, UK
REVIEW RETURNED	18-Sep-2019

GENERAL COMMENTS	This paper describes a cohort of 1000 families of children with intellectual disability, that took part in a study of family wellbeing. The cohort has strengths due to its size and includes cross-sectional and longitudinal data. While the study overall is important, this specific manuscript could be tightened to make sure that it is describing the cohort rather than the study as a whole. I have some comments that the authors may like to consider when revising this manuscript. Abstract: - I think that the information in the 'Participants' and 'Findings to date' subsections of the abstract should be switched. Currently, the findings to date section includes a characterization of the sample, and the participants section contains information about the data that has been collected. I think that this should be reversed. Strengths and limitations: - The strengths and limitations section makes reference to the 'study' in addition to the 'cohort'. Given that this manuscript is a cohort profile, the strengths/limitations could be tightened and more specifically describe the value of the cohort, for example, by stating that it includes people that cover the whole range of ID (from mild to severe). Cohort description: - On p 6. Lines 4-6, the authors indicate that the sample includes 1000 families of children with ID. I wonder if it is worth specifying at this point that the families were all living in the UK at the time of recruitment.- - On p. 6 lines 35-36, the authors state that telephone interviews were completed by 649 primary caregivers. It may be helpful to indicate whether the characteristics of these families differ in any way to the characteristics of families who chose not to participate in the telephone interviews. I appreciate that this is in the strengths and limitations section, but it could be presented more succinctly.
---

	- Table 2 (presented on p. 11) describes the measures used within the study and indicates that they are measured at T1 and T2. Does this mean that these measures were collected during wave 1 and wave 2? - I would like to see a more detailed description of the child with ID. For example, the mean scores (and range) on the Vinelands that were collected and whether the authors collected any information about cognitive assessments that families could report, e.g., based on previous diagnostic assessments. - In the description of the primary caregiver and family, it would be helpful to know how many siblings were in each family and whether the siblings had any diagnoses, such as ID or ASD. - In the cohort description, I would also like to know whether the authors collected information on other genetic conditions typically associated with ID, aside from autism and Down Syndrome (for example, Fragile X, tuberous sclerosis, Williams Syndrome, Cornelia De Lange etc.). Similarly, did the authors collect any information about medical comorbidities (e.g., epilepsy, cerebral palsy), which may affect family wellbeing over and above the ID on its own? It would be worthwhile to present this information in the description of the cohort. - The authors have included information about the proportion of families living in the most deprived areas of the UK. It would be helpful to see where the participating families came from across the UK, perhaps using a visual? Findings to Date: - Much of the information in the 'Findings to Date' section belongs with the cohort description, as the authors are referring to the characterisation of the sample. Have there been any key findings or publications related to family wellbeing that have come from this sample? - The families were asked to describe the severity of their child's ID. It would be helpful to know how this was operationalised for the families. Future Directions: - The authors could provide more specific details about the questions that they hope to answer using this cohort and the rationale for future work. Strengths and Limitations: - In my opinion, the way that the comparisons of the characteristics of the families that did and did not take part in the telephone interviews is not very meaningful. Simply stating that one group had a higher mean score than the other, without indicating whether this difference was significant, nor an effect size or confidence intervals, does not tell us very much. I think this section could be reworked and presented more succinctly.
--	--

REVIEWER	mitchell Schertz Neurodevelopmental service Meuhedet north
REVIEW RETURNED	26-Oct-2019

GENERAL COMMENTS	item #13 - i did not assess this point Title: Cohort Profile: The 1,000 Families Study, a UK multi-wave cohort investigating the wellbeing of families of children with intellectual disabilities. Summary
---

This manuscript reported on an ongoing cohort study of over 1000 families of children with intellectual disabilities ages 4-15 years. The report describes the methods of the study including detailed listing of measures to be obtained. It further provides descriptive information about the families. Finally, it reports that next stage (Wave 2) data collection is ongoing.

General Comment:

I commend the authors for addressing the very important issue of treatment of children with ID as well as the comprehensiveness of the measures to be undertaken. However, the data as presented seems limited in order to make any clear assessment of the value of the study.

In Table 1. SES and demographic data on Wave 1 information is presented. In Table 2, measures used within the study are presented as well. Notably, for most of the measures presented, the majority (> 50%, based on my calculations) were reported as being already completed by the parent. Yet very limited amounts of this were presented (see comment #5 below and page 19 of the manuscript). It is not clear to this reviewer why this data was not presented, as this precludes one's ability to adequately assess this manuscript. If more data is not available, then it is not stated what is the value and justification of publishing this manuscript at this stage, as opposed to waiting for completion of the next stage of data collection.

To follow are a few comments on the manuscript as presented.

Comment #1:

Page 3, Line 246

"Prevalence studies internationally suggest that approximately 2-3% of children and adolescents have an ID[2-3]."

Unclear why reference #3 by Totsika, Hastings Emerson et al., is used as this is not a direct study of prevalence of ID and indeed mentions the study as a secondary analysis of other data. More recent references exist. I note for example two:

Platt, Jonathan M., et al. "Intellectual disability and mental disorders in a US population representative sample of adolescents." *Psychological medicine* 49.6 (2019): 952-961.

Anderson, Lynda Lahti, et al. "A systematic review of US studies on the prevalence of intellectual or developmental disabilities since 2000." *Intellectual and Developmental Disabilities* 57.5 (2019): 421-438.

Comment #2:

Page 5, Line 22

"Study aims...The main aims of the 1,000 Families Study are:"

Besides noting the goal to "explore correlates of...", It might be valuable to examine specific questions that the literature currently does not address with focused hypothesis that can be addressed.

Thus, for the example the authors in the introduction note the increased risk of both maternal and paternal psychiatric morbidity in those having a child with ID. What further information might the current study provide in this respect that justifies further study.

Comment #3; Page 17, Line 33-47

"This study is, as far as we are aware, the largest cohort of families with a child with ID in the

UK and will make an important contribution to the research on the wellbeing of children with ID and their families. Particularly important is the longitudinal design which currently includes a

second wave of data collection that is underway. Selection of measures allows for some comparison with population-representative samples in the UK (e.g. the Millennium Cohort Study[24]).”

It would be helpful to make comparison to other large cohort studies from other countries.

Comment #4; Page 17, Line 56

“Instead, information on the child’s diagnoses is based solely on parental report”

Given the large amount of information obtained from the parents, why not simply ask the parent what educational setting the child is attending – as this would provide at least a preliminary correlation with the function of the child.

Comment #5; Page 19, Line 3-31

“Children of caregivers who did not take part in the telephone interview had higher total SDQ scores with a mean score of 18.3 for the children whose primary caregivers did not take part in the telephone interview compared to a mean score of 16.8 for those who did. The child with ID having multiple diagnoses, however, did not decrease the chances that a primary caregiver would complete a telephone interview. For example, those children with ID whose primary caregivers completed a telephone interview were more likely to have a coexisting physical health problem (such as epilepsy, mobility problems, hearing or visual problems). The children whose primary caregivers completed a telephone interview had a mean number of health problems of 1.70 compared to 1.06 for the children with ID whose primary caregivers did not complete a telephone interview. Additionally, the children whose primary caregivers completed a telephone interview were more likely to have a greater number of diagnostic labels, with a mean number of 2.18 compared to 1.45 for the children with ID whose primary caregivers did not complete a telephone interview.”

From my reading of the manuscript, the above quoted paragraph contains the only information regarding results of measures – and specifically comparing parents who completed the phone interview vs. those that did not. Why is this information not located on page 15 under the heading “FINDINGS TO DATE”?

Clarification of this point is important to allow the reviewer to understand why only this information was presented and not others. More complete statistical data, such as standard deviation, would also be helpful.

END OF REVIEW

VERSION 1 – AUTHOR RESPONSE

Reviewer #1 comments:

1.1 “This paper describes a cohort of 1000 families of children with intellectual disability, that took part in a study of family wellbeing. The cohort has strengths due to its size and includes cross-sectional and longitudinal data. While the study overall is important, this specific manuscript could be tightened to make sure that it is describing the cohort rather than the study as a whole”.

RESPONSE

Thank you for your positive comments. We have made edits throughout the manuscript to ensure the focus is on describing the cohort, and to provide further information on the cohort to highlight the strengths outlined. Edits in response to the other reviewers’ comments have also helped to achieve this.

1.2 “I think that the information in the 'Participants' and 'Findings to date' subsections of the abstract should be switched. Currently, the findings to date section includes a characterization of the sample, and the participants section contains information about the data that has been collected. I think that this should be reversed.”

RESPONSE

This order of information has been reversed as suggested ('Abstract', p. 2).

1.3 “The strengths and limitations section makes reference to the 'study' in addition to the 'cohort'. Given that this manuscript is a cohort profile, the strengths/limitations could be tightened and more specifically describe the value of the cohort, for example, by stating that it includes people that cover the whole range of ID (from mild to severe).”

RESPONSE

This section has been reviewed to clarify where ‘cohort’ and ‘study’ are the most appropriate terms to use (article summary, p. 3). Further points relating to the cohort, such as the fact it includes children from mild to severe ID ('Wave 1 Cohort Description', p. 8), have been included.

1.4 “On p 6. Lines 4-6, the authors indicate that the sample includes 1000 families of children with ID. I wonder if it is worth specifying at this point that the families were all living in the UK at the time of recruitment.”

RESPONSE

This information has been added.

1.5 “On p. 6 lines 35-36, the authors state that telephone interviews were completed by 649 primary caregivers. It may be helpful to indicate whether the characteristics of these families differ in any way to the characteristics of families who chose not to participate in the telephone interviews. I appreciate that this is in the strengths and limitations section, but it could be presented more succinctly.”

RESPONSE

We have added a sentence at this point in the paper (‘Cohort Description’, p. 6) to highlight where the information comparing the interview cohort and the overall cohort can be found. More complete statistical information has then been added in the findings to date section (pp. 18-20), and this is also summarized succinctly in the strengths and limitations (pp. 22-23).

1.6 “Table 2 (presented on p. 11) describes the measures used within the study and indicates that they are measured at T1 and T2. Does this mean that these measures were collected during wave 1 and wave 2?”

RESPONSE

Yes, these measures were collected during Wave 1 and Wave 2 – the table has been updated to refer to W1 and W2 rather than T1 and T2, and these are defined in the note below the tables (see Table 3, pp.14-17).

1.7 “I would like to see a more detailed description of the child with ID. For example, the mean scores (and range) on the Vinelands that were collected and whether the authors collected any information about cognitive assessments that families could report, e.g., based on previous diagnostic assessments.”

RESPONSE

Further information on the child, including mean scores and range on the Vineland as well as medical comorbidities, have been included in Table 1 (pp.11-12), as well as summarized in the text (such as Wave 1 Cohort description, pp. 7-9, and Findings to Date, pp.18-20). No cognitive assessments were conducted, and this is now discussed in the strengths and limitations (p.22).

1.8 “In the description of the primary caregiver and family, it would be helpful to know how many siblings were in each family and whether the siblings had any diagnoses, such as ID or ASD.”

RESPONSE

This information has been added to Table 2 (pp.13-14), which now focuses on the socio-demographic information for the caregiver and family.

1.9 “In the cohort description, I would also like to know whether the authors collected information on other genetic conditions typically associated with ID, aside from autism and Down Syndrome (for example, Fragile X, tuberous sclerosis, Williams Syndrome, Cornelia De Lange etc.). Similarly, did the authors collect any information about medical comorbidities (e.g., epilepsy, cerebral palsy), which may affect family wellbeing over and above the ID on its own? It would be worthwhile to present this information in the description of the cohort.”

RESPONSE

Information on the child (including diagnoses and medical co-existing conditions) has now been added to the text (p. 8) and Table 1 (pp.11-12) to provide a more complete description of the children and their families.

1.10 “The authors have included information about the proportion of families living in the most deprived areas of the UK. It would be helpful to see where the participating families came from across the UK, perhaps using a visual?”

RESPONSE

Location by UK country is now reported for families in Table 2 (pp. 13-14) and in the text (Wave 1 Cohort description, p. 8). We have chosen not to add a visual so as not to take up unnecessary space.

1.11 “Much of the information in the ‘Findings to Date’ section belongs with the cohort description, as the authors are referring to the characterisation of the sample. Have there been any key findings or publications related to family wellbeing that have come from this sample?”

RESPONSE

Our understanding of a Cohort Profile paper was that we should focus on describing the achieved cohort. No findings have been reported to date, and findings addressing the research questions will be reported in future papers. Any future publications (none currently exist) will cite the cohort paper. Therefore, this manuscript focuses on the cohort description as well as the background to the overall study.

1.12 “The families were asked to describe the severity of their child’s ID. It would be helpful to know how this was operationalised for the families.”

RESPONSE

A description of how severity of ID was defined for parent report has been added to the following section: Wave 1 Cohort Description (p. 8).

1.13 “The authors could provide more specific details about the questions that they hope to answer using this cohort and the rationale for future work.”

RESPONSE

Further details have been added to Future Directions (p. 21) to clarify some of the questions that will be explored with this cohort.

1.14 “In my opinion, the way that the comparisons of the characteristics of the families that did and did not take part in the telephone interviews is not very meaningful. Simply stating that one group had a higher mean score than the other, without indicating whether this difference was significant, nor an effect size or confidence intervals, does not tell us very much. I think this section could be reworked and presented more succinctly.”

RESPONSE

The section on characteristics of the families that did and did not take part in the telephone interview has been reworked, and now includes additional statistics allowing meaningful comparison. The findings are discussed in Findings to Date (pp.18-20) and are summarized succinctly in Strengths and Limitations (pp. 22-23).

Reviewer #2 comments:

2.1 “However, the data as presented seems limited in order to make any clear assessment of the value of the study. In Table 1. SES and demographic data on Wave 1 information is presented. In Table 2, measures used within the study are presented as well. Notably, for most of the measures presented, the majority (> 50%, based on my calculations) were reported as being already completed by the parent. Yet very limited amounts of this were presented (see comment #5 below and page 19 of the manuscript). It is not clear to this reviewer why this data was not presented, as this precludes one’s ability to adequately assess this manuscript. If more data is not available, then it is not stated what is the value and justification of publishing this manuscript at this stage, as opposed to waiting for completion of the next stage of data collection.”

RESPONSE

See response to Reviewer 1 1.11. Also, in responding to points from both reviewers, more information about the cohort has been added to the paper.

2.2 “Unclear why reference #3 by Totsika, Hastings Emerson et al., is used as this is not a direct study of prevalence of ID and indeed mentions the study as a secondary analysis of other data More recent references exist. I note for example two: Platt, Jonathan M., et al. "Intellectual disability and mental disorders in a US population representative sample of adolescents." *Psychological medicine* 49.6 (2019): 952-961. 2 Anderson, Lynda Lahti, et al. "A systematic review of US studies on the prevalence of intellectual or developmental disabilities since 2000." *Intellectual and Developmental Disabilities* 57.5 (2019): 421-438.”

RESPONSE

The suggested references have been added and the approximate prevalence rate reported has been revised (Introduction, p. 3).

2.3 “Besides noting the goal to “explore correlates of...”, It might be valuable to examine specific questions that the literature currently does not address with focused hypothesis that can be addressed. Thus, for the example the authors in the introduction note the increased risk of both maternal and paternal psychiatric morbidity in those having a child with ID. What further information might the current study provide in this respect that justifies further study.”

RESPONSE

The Future Directions section has been updated to provide a more detailed summary of future areas of research using data from this cohort (p. 21). However, given our understanding of the purpose of a cohort profile, we have not made further changes.

2.4

“It would be helpful to make comparison to other large cohort studies from other countries.”

RESPONSE

Two example cohort studies from other countries have now been referenced in this section to acknowledge the international context (Strengths and limitations, p. 21). However, our main point here was about the selection of measures that would provide opportunities for comparison with other UK datasets and not comparison with other international datasets.

2.5 “Given the large amount of information obtained from the parents, why not simply ask the parent what educational setting the child is attending – as this would provide at least a preliminary correlation with the function of the child.”

RESPONSE

Educational setting was included in Table 1. This has now also been added to the text alongside a summary of parent-reported intellectual disability severity and VABS scores for those who completed the telephone interview (Wave 1 Cohort Description, p. 8). Limitations of this approach are discussed in the Strengths and Limitations (p. 22).

2.6 “From my reading of the manuscript, the above quoted paragraph contains the only information regarding results of measures – and specifically comparing parents who completed the phone interview vs. those that did not. Why is this information not located on page 15 under the heading “FINDINGS TO DATE”? Clarification of this point is important to allow the reviewer to understand why only this information was presented and not others. More complete statistical data, such as standard deviation, would also be helpful.”

RESPONSE

As requested, we have moved this information to ‘Findings to date’ (pp. 18-20) and we added further information to provide a clearer picture of those who did, and did not, take part in the telephone interview.

CONCLUDING COMMENTS

We have included a ‘marked copy’ with changes marked clearly as well as a ‘clean’ version of the updated manuscript. We thank the two reviewers for their feedback and consideration of this manuscript and we look forward to seeing our paper published with *BMJ Open*. Whilst making revisions, we took the opportunity to carry out a careful check of the data, and have included minor updated numbers in the tables and text where needed.

VERSION 2 – REVIEW

REVIEWER	Lauren Taylor King's College London, Institute of Psychiatry, Psychology and Neuroscience, UK
REVIEW RETURNED	18-Dec-2019

GENERAL COMMENTS	Thank you for the opportunity to review the revised manuscript titled, "Cohort Profile: the 1,000 Families Study, a UK multi-wave cohort investigating the wellbeing of families of children with intellectual disabilities." This manuscript describes a cohort of 1,000 families of children with an intellectual disabilities, who are taking part in a study designed to investigate wellbeing in these families. The authors were very responsive to prior review comments and have subsequently made substantial changes in
---

	response to the reviewers comments. I have only very minor suggestions for further revision. On the whole the revised manuscript more clearly distinguished between the 1,000 Families Study and the description of the cohort participating in the study. The additional detail about the child with intellectual disability also helped to contextualise the information that was presented in the manuscript. My only suggestion is that there is a section heading for STUDY DESCRIPTION (or something similar), with following sections for Study aims, Wave 1 data collection, Wave 2 data collection, Study funding, measures, PPI & Ethics, because these are all general aspects of the 1,000 Families Study. This would move the heading for COHORT DESCRIPTION further down the paper and would more clearly link the description of the cohort with information about the inclusion/exclusion criteria and cohort description/sample characteristics. It may also be helpful to open this section with a statement that the focus of the rest of the manuscript is a description of the cohort participating in the 1,000 Families Study.
--	--

REVIEWER	mitchell Schertz Neurodevelopmental service Meuhedet north
REVIEW RETURNED	05-Dec-2019

GENERAL COMMENTS	General Comment: In reviewing the revised manuscript, there is some clarification on the part of the authors that they see the aim of the current manuscript as a presentation of the cohort and not a presentation of the data collected from the cohort participants (see their response to comment 2.3). Yet, this is NOT explicitly stated neither in the abstract under Goal/Purpose nor in the manuscript. It would provide clarification to the reader if the stated aims of this being a description of the cohort appear in both the abstract and in the manuscript. What the goals of the cohort study should be mentioned in a separate context. I have referenced for example 2 articles of cohort descriptions that also use this suggested format Justice, A. C., Dombrowski, E., Conigliaro, J., Fultz, S. L., Gibson, D., Madenwald, T., ... & RodriguezBarradas, M. C. (2006). Veterans aging cohort study (VACS): overview and description. Medical care, 44(8 Suppl 2), S13. Casasnovas, J. A., Alcaide, V., Civeira, F., Guallar, E., Ibañez, B., Borreguero, J. J., ... & Pocovi, M. (2012). Aragon workers' health study—design and cohort description. BMC cardiovascular disorders, 12(1), 45. I have responded in bold to the Authors Revisions noted below. There is format problem in the references section in regard to reference #2 that should be corrected.  _____ Review response to Authors' Revisions Reviewer #2 comments: 2.1 "However, the data as presented seems limited in order to make any clear assessment of the value of the study. In Table 1. SES and demographic data on Wave 1 information is presented. In Table 2, measures used within the study are presented as well. Notably, for most of the measures presented, the majority (> 50%, based on my
--

	calculations) were reported as being already completed by the parent. Yet very limited amounts of this were presented (see comment #5 below and page 19 of the manuscript). It is not clear to this reviewer why this data was not presented, as this precludes one's ability to adequately assess this manuscript. If more data is not available, then it is not stated what is the value and justification of publishing this manuscript at this stage, as opposed to waiting for completion of the next stage of data collection." RESPONSE See response to Reviewer 1 1.11. Also, in responding to points from both reviewers, more information about the cohort has been added to the paper. Reviewer: OK 2.2 "Unclear why reference #3 by Totsika, Hastings Emerson et al., is used as this is not a direct study of prevalence of ID and indeed mentions the study as a secondary analysis of other data. More recent references exist. I note for example two: Platt, Jonathan M., et al. "Intellectual disability and mental disorders in a US population representative sample of adolescents." Psychological medicine 49.6 (2019): 952-961. 2 Anderson, Lynda Lahti, et al. "A systematic review of US studies on the prevalence of intellectual or developmental disabilities since 2000." Intellectual and Developmental Disabilities 57.5 (2019): 421-438." RESPONSE The suggested references have been added and the approximate prevalence rate reported has been revised (Introduction, p. 3). Reviewer: OK 2.3 "Besides noting the goal to "explore correlates of...", It might be valuable to examine specific questions that the literature currently does not address with focused hypothesis that can be addressed. Thus, for the example the authors in the introduction note the increased risk of both maternal and paternal psychiatric morbidity in those having a child with ID. What further information might the current study provide in this respect that justifies further study." RESPONSE The Future Directions section has been updated to provide a more detailed summary of future areas of research using data from this cohort (p. 21). However, given our understanding of the purpose of a cohort profile, we have not made further changes. Reviewer: OK 2.4 "It would be helpful to make comparison to other large cohort studies from other countries." RESPONSE Two example cohort studies from other countries have now been referenced in this section to acknowledge the international context (Strengths and limitations, p. 21). However, our main point here was about the selection of measures that would provide opportunities for comparison with other UK datasets and not comparison with other international datasets. Reviewer: Unfortunately, I do not agree. If the goal of this study is to describe a cohort (see above highlighted in yellow), then a comparison with other cohort studies is in order.
--	--

	2.5 “Given the large amount of information obtained from the parents, why not simply ask the parent what educational setting the child is attending – as this would provide at least a preliminary correlation with the function of the child.” RESPONSE Educational setting was included in Table 1. This has now also been added to the text alongside a summary of parent-reported intellectual disability severity and VABS scores for those who completed the telephone interview (Wave 1 Cohort Description, p. 8). Limitations of this approach are discussed in the Strengths and Limitations (p. 22). Reviewer: OK 2.6 “From my reading of the manuscript, the above quoted paragraph contains the only information regarding results of measures – and specifically comparing parents who completed the phone interview vs. those that did not. Why is this information not located on page 15 under the heading “FINDINGS TO DATE”? Clarification of this point is important to allow the reviewer to understand why only this information was presented and not others. More complete statistical data, such as standard deviation, would also be helpful.” RESPONSE As requested, we have moved this information to ‘Findings to date’ (pp. 18-20) and we added further information to provide a clearer picture of those who did, and did not, take part in the telephone interview. Reviewer: OK
--	---

VERSION 2 – AUTHOR RESPONSE

Reviewer #1 comments:

1.1 My only suggestion is that there is a section heading for STUDY DESCRIPTION (or something similar), with following sections for Study aims, Wave 1 data collection, Wave 2 data collection, Study funding, measures, PPI & Ethics, because these are all general aspects of the 1,000 Families Study. This would move the heading for COHORT DESCRIPTION further down the paper and would more clearly link the description of the cohort with information about the inclusion/exclusion criteria and cohort description/sample characteristics. It may also be helpful to open this section with a statement that the focus of the rest of the manuscript is a description of the cohort participating in the 1,000 Families Study.

RESPONSE

The two subtitles have been added as requested:

STUDY DESCRIPTION (p.5) and COHORT DESCRIPTION (p.18) along with a sentence introducing the latter section (p.18)

Reviewer #2 comments:

2.1 It would provide clarification to the reader if the stated aims of this being a description of the cohort appear in both the abstract and in the manuscript. What the goals of the cohort study should be mentioned in a separate context.

RESPONSE

The abstract has been edited to clarify that the manuscript provides a description of the cohort, with future articles exploring and presenting the findings from further analyses (p.2) as well as in the manuscript (p.5).

2.2 There is format problem in the references section in regard to reference #2 that should be corrected.

RESPONSE

This has been corrected (p.26).

2.3 “It would be helpful to make comparison to other large cohort studies from other countries.”

RESPONSE

Two example cohort studies from other countries have now been referenced in this section to acknowledge the international context (Strengths and limitations, p. 21). However, our main point here was about the selection of measures that would provide opportunities for comparison with other UK datasets and not comparison with other international datasets.

Reviewer: Unfortunately, I do not agree. If the goal of this study is to describe a cohort (see above highlighted in yellow), then a comparison with other cohort studies is in order.”

RESPONSE

We have provided some further descriptive information about the Adolescents and Adults with Autism Study (p.21-22) and compared their study design to our own.

CONCLUDING COMMENTS

We have included a ‘marked copy’ with changes marked clearly as well as a ‘clean’ version of the updated manuscript. We thank the two reviewers for their feedback and consideration of this manuscript and we look forward to seeing our paper published with BMJ Open.